# Effect of MBF-20 Interlayer on the Microstructure and Corrosion Behaviour of Inconel 625 Super Alloy after Diffusion Brazing

**DOI:** 10.3390/ma16145072

**Published:** 2023-07-18

**Authors:** Alireza Doroudi, Hamid Omidvar, Ali Dastgheib, Mohammad Khorram, Armin Rajabi, Amir Hossein Baghdadi, Mariyam Jameelah Ghazali

**Affiliations:** 1Department of Mining and Metallurgical Engineering, Amirkabir University of Technology (Tehran Polytechnic), Tehran 15875-4413, Iran; doroudi.alireza@aut.ac.ir (A.D.); omidvar@aut.ac.ir (H.O.); 2Corrosion Engineering and Material Protection Group, Bandar Abbas Campus, Amirkabir University of Technology (Tehran Polytechnic), Tehran 15875-4413, Iran; dastgheibali1993@gmail.com (A.D.); mohammad.khorram@aut.ac.ir (M.K.); 3Department of Mechanical and Manufacturing Engineering, Faculty of Engineering and Built Environment, University Kebangsaan Malaysia (UKM), Bangi 43600, Selangor, Malaysia; baghdadi.amirhossein@gmail.com

**Keywords:** Inconel 625, MBF-20, diffusion brazing, boride precipitation, corrosion

## Abstract

The joining zone includes three main parts, which comprise an isothermal solidification zone (ISZ), the athermal solidification zone (ASZ), and a diffusion affected zone (DAZ). Field emission scanning electron microscopy (FESEM) was used here to observe the microstructure equipped with ultra-thin window energy dispersive X-ray spectrometer (EDS) system. Additionally, electrochemical impedance spectroscopy (EIS) and cyclic potentiodynamic polarization tests were conducted to evaluate the effect of the DB process on the corrosion resistance of the Inconel 625 superalloy. In the bonding time period, some Mo- and Cr-rich boride precipitations and Ni-rich γ-solid solution phases with hardened alloy elements, such as Mo and Cr, formed in DAZ and ASZ, respectively, because of the inter-diffusion of melting point depressants (MPD). Moreover, during cooling cycles, Ni-Cr-B, Ni-Mo-B, Ni-Si-B, and Ni-Si phase compounds were formed in the ASZ area at 1110–850 °C. The DAZ area developed by borides compound with cubic, needle, and grain boundary morphologies. The corrosion tests indicated that the DB process led to a reduction in the passive region and increased the sensitivity to pitting corrosion.

## 1. Introduction

Inconel 625 is a chromium-rich, nickel-based superalloy that has outstanding mechanical and corrosion resistance properties [1]. The mechanism of strengthening of Inconel 625 is solid–solution hardening, which is due to the presence of molybdenum and niobium in the Ni-Cr matrix [2]. High-temperature components need to be repaired or replaced in the first place due to harsh operating condition [3]. Advancing repair methods, such as diffusion brazing (DB), should be used to decrease the numerous defects of the material in high-temperature cracking. Generally, the DB process consists of three steps: (a) dissolution of base metal, (b) isothermal solidification, and (c) solid-state homogenisation [4]. In the binary systems, the formation of isothermal solidification via a solid-state homogenisation leads to the creation of a joining zone with an identical chemical composition as the base metal. To perform the DB process, usually an interlayer containing melting point depressant elements (MPD) is used [5]. By decreasing the concentration of MPD elements in the interlayer, the melting point is increased and the liquid phase formed by melting the interlayer is solidified at a constant temperature, which is called the isothermal solidification zone (ISZ) [6].

A study on the diffusion brazing of an Ni-based superalloy with a B-rich interlayer reported that the microstructure of the joint after partial isothermal solidification consists of three main sub-regions [7,8,9,10,11] as follow:

The isothermal solidification zone (ISZ) does not have any free-boride intermetallic phases and contains Ni-rich γ solid–solution phase, which is formed during the diffusion of MPD elements and the dissolution of alloying BM elements in the joining zone. In the athermal solidification zone (ASZ), during cooling in the vacuum furnace, B and Si diffuse back to the liquid phase, which is highly concentrated by hardening elements from the BM due to insufficient diffusion time. As a result, athermal solidification bonding is formed from the remaining liquid in the centreline. This process can lead to the formation of hard and brittle phases. In the diffusion affected zone (DAZ), B is diffused from the interlayer material into the BM and hard boride phases are formed due to the chemical reaction between B and alloying elements, such as Cr, Mo, and Ni. This area contains boride precipitates.

Studies on the diffusion bonding of Inconel 625 superalloy reported that the joining zone includes Cr- and Mo-rich boride, Ni-rich boride and Ni-rich silicide precipitates [12]. In another work, eutectic ternary phases, such as $K\;/Ni_{3}B\;/Ni_{6}Si_{2}B$, were formed in the centreline of the joint during the solidification of Inconel 718 [13]. $K+Ni_{3}B+CrB$ ternary eutectic phases and $K+Ni_{3}Si$ binary eutectic phases were formed in the joining zone for Inconel 738-LC [14]. The formation of boride precipitates in DAZ are accelerated by increasing the formation of boride phases in the substrate [15,16]. Moreover, Yuan et al. [17] found that Si and B are responsible for two different structures during bonding via DB for duplex stainless steel. In addition, [18] used to DSC technique for defined isothermal solidification rate on Inconel 625 TLP bonded. They measured solidification enthalpy from liquid eutectic in the centreline during the cooling cycle, which showed that one diffusional solidification occurred during the primary melting of the interlayer.

Inconel 625 is widely used in applications where corrosion and creep resistance is required [19], because it has high pitting corrosion resistance [20]. Kangazian et al. [21] addressed the formation of carbides, borides, and secondary phases because Inconel 625 consists of Cr, Mo, and Ni; after welding, these phases can be formed, depending on the holding time and temperature. Given that borides and carbides have higher concentrations of chromium and molybdenum than the BM, these phases are responsible for the deterioration of corrosion resistance and physical properties. The corrosion rate of the alloy increases with the decreasing concentration of Mo and Cr due to the increase in the cathode reaction rate (i.e., $H^{+}$ ion reduction) [22,23]. The emergence of artistic phases leads to the depletion of Cr in diffuse brazed areas; this phenomenon is undesirable and weakens the corrosion behaviour of the joints, leading to the same width of Cr-depleted zone and ASZ [24].

According to previous research [6], the corrosion resistance increased with decreasing ASZ width. Mirzaei et al. [25] investigated the effect of TLP process time of 304 stainless steel by using the MBF-20 interlayer and its mechanical properties and corrosion behaviour in 3.5 wt% NaCl solution. It was reported that there is an optimal limit on the amount of corrosion resistance of the joint region. So that with increasing bonding time, the corrosion current density (i_corr_) first decreases and then increases. This was due to a change in the amount of precipitate in the joint region. Another study by Mosallaee et al. [26] on the effect of the TLP of DSS-2205/Ni–P/DSS-2205 specimens by BNi-6 interlayer, demonstrates that the prior to the completion of the isothermal solidification region, the joining region was the weakest region for pitting corrosion, and the pitting attack occurred close to the rich eutectic phases (Mo, Cr, and P) in the joining region centreline. After completing the IS, the DAZ acted as the most susceptible site to pitting corrosion and the pits selectively occurred close to (Mo–P) rich intermetallic phases in the DAZ. In this work, the microstructure of the joining zone for the Inconel 625 superalloy after diffusion brazing (1120 °C for 30 min) was studied. To the best of our knowledge, no research has reported on microstructure analysis of this alloy and the processing condition mentioned. The formation of borides and silicide in the joining zone was observed, and a correlation between microstructure, corrosion behaviour, and bonding temperature was proposed. Results show that phase compounds were formed within a range of temperature from 1100 °C to 850 °C after bonding, and an ASZ eutectic structure was formed during cooling cycles.

## 2. Materials and Methods

Table 1 shows the chemical composition of Inconel 625 and MBF-20 as base metal (BM) and interlayer for diffusion bonding, respectively. The samples were cut for brazing by electro-discharge machine (EDM) in 10 mm × 10 mm × 5 mm pieces. The samples were sanded to 1200 gravel by using silicon carbide sandpaper to remove surface contaminants. For DB, an MBF-20 interlayer with a thickness of 75 μm was used. The chemical composition of the interlayer is listed in Table 1. The samples were placed in acetone for 30 min to remove organic compounds on the surface. There is used as a fixture (Figure 1), because the interlayer must be held between two samples, and fixed. Furthermore, due to the high-temperature process, the fixture was manufactured of stainless steel in an attempt to prevent deformation during the bonding process. So, a homemade stainless steel fixture (Figure 1) was used to grab the samples inside the furnace. The samples and the sample holder were placed in the vacuum furnace at 10^−5^ mbar pressure and held at 1120 °C for 30 min. The heating rate of the furnace was 15 °C per minute, and the samples were cooled inside the furnace in vacuum.

After the DB process, the samples were cut perpendicular to the joining surface by using EDM. The microstructure of the joining zone was studied by the following protocol: (1) Oxalic acid solution containing 10% oxalic acid and 90% water was used as an etchant solution and etching was conducted at 6 V to prepare the surface for the samples. (2) Olympus^®^ inverted-reflected light microscopy (Olympus GX41-Japan), (Philips XL30^®^ scan electron microscopy (SEM), and Tescan Mira3 XMU^®^ field emission scanning electron microscopy (FESEM) were used to observe the microstructure. (3) Image J 1.44p^®^ software was used to measure the average size of the areas in the joining zone and the size of precipitates based on the method recommended by [27]. (4) An ultra-thin window energy dispersive X-ray spectrometer (EDS) armed FESEM (15 kV accelerating voltage) was utilised to determine the chemical composition.

EIS and polarisation tests were conducted in order to understand the corrosion behaviour of the joints after the DB process. Before conducting the tests, each specimen was electrically connected to a conductive wire and the specimens were degreased with acetone, rinsed with deionized water, and dried with warm air, then covered with a beeswax-colophony blend (ratio 3 beeswax to 1 colophony) so that a 0.1 cm^2^ area of each specimen was exposed to the 3.5 wt% NaCl as an electrolyte. A standard three-electrode workstation was used, and the samples were connected to the working electrode terminal. Ag /AgCl  (3MKCl) electrode and platinum wire were used as reference electrode and counter electrode, respectively. The model of the three-electrode workstation was AutolabPGSTAT Model 302N with FRA module and NOVA 2.1.4 software. Figure 2 shows the schematic of the electrochemical test. Electrochemical tests were performed after immersing the specimens for an hour and the open circuit potential became steady (dE/dt < 10^−6^ V).

EIS test was carried out within the 10 *mHz*–100 *kHz* frequency range and 10 mV perturbation amplitude. Furthermore, ZView 3.1c software was utilised to obtain the equivalent circuit and fit the EIS test results. The working electrode potential was swept from −250 *mV* versus Ag/AgCl. The scanning rate was 1 $mVs^{-1}$. The potential was then inverted to obtain a well-defined loop of hysteresis. Each test was performed twice to ensure accuracy.

## 3. Results

### 3.1. Microstructure Characterization

After melting the interlayer at bonding temperature, the chemical composition of the liquid phase was reformed by several mass transportation phenomena: the dissolution of the BM and inter-diffusion between BM and interlayer, leading to the formation of boride and nickel silicide precipitates. Figure 3a shows the microstructure of the joining zone at 1120 °C for 30 min. The joining zone contained ISZ, ASZ, and DAZ. In a previous study, Ni_3_Mo was formed in the grain (the δ phase) and titanium carbide was formed on the grain boundary [28,29]. The EDS line passing across the joint Figure 3b showed the distribution of MPD elements and BMs. 

#### 3.1.1. Isothermal Solidification Zone

The EDS analysis of point ‘A’ Figure 3a and Table 2 showed that the ISZ consisted of Ni-rich γ-phase containing Mo and Cr because of the dissolution of hardening, alloying elements from the BM in the joint (as shown by the EDS-line analysis in Figure 3b). This region was formed through a solidification mechanism similar to isothermal solidification. During IS, the chemical composition of liquid between the interlayer and BM changed due to the inter-diffusion of the alloying elements, that is, the diffusion of B into the BM is able to control the IS mechanism [15]; thus, IS width depends on the chemical composition and bonding time. The distribution coefficient (K) of the elements has an important effect on the completion of ISZ. $K=C_{S}^{i}/C_{L}^{i}$ is the distribution coefficient (K) of an element, where $C_{S}$ and $C_{L}$ are the concentration of the solute in the solid and melting material, respectively. So, if the K of elements is less than one, they tend to stay in the melt, and melt is enriched by them. Therefore, for K>1, the concentration of these elements in front of interface can increase due to the rejection of the alloying element from the solid [30].

Two phenomena associated with the increase in the liquid temperature of the liquid phase are as follows [13]; firstly, B depletion from the liquid phase (K<1); and secondly, the concentration of the liquid phase by the BM’s hardening alloying elements (i.e., Mo, Cr, and Nb) with a distribution coefficient of more than one (K>1).

#### 3.1.2. Athermal Solidification Zone

When the bonding time is insufficient to allow the diffusion of MPD atoms into the BM, a eutectic structure was formed in the centreline during cooling. According to Figure 3a, the eutectic phase was formed in the middle of the joint. The EDS analysis of points ‘B–F’ in Table 2 showed that this area was made of boride and silicide. Some researchers [31,32,33,34,35] highlighted that eutectic phases were formed in the ASZ. During IS from the initial γ -phase, B is pushed back into the liquid phase because of the poor solubility of B into Ni and its lower distribution coefficient than unity (K < 1). Based on the solubility of Cr and Mo in the joint centreline, B is trapped in the liquid phase and forms boride precipitates. The EDS analysis showed that B and Si were concentrated at the joint, leading to a decrease in the melting point of the liquid phase; the melting point became less than the bonding temperature in the centreline. Ghasemi et al. [31] used Cr-B and Mo-B binary diagrams and Thermo Calc software and reported that the concentrations of B and Si at 1120 °C in Cr and Mo are equal to 0.35 and 0.0089 at %, respectively. Another study on Ni-B and Ni-Si binary diagram showed that the concentrations of B and Si are equal to 0.2 and 14 at %, respectively [5].

Figure 3a shows the specified area of ASZ, which presented the ASZ chemical composition and the phases formed. According to the EDS analysis in Table 2 and the phase diagram, the following phases were formed:(1)The Ni-Cr-B phase might be formed in the bonding zone due to the dissolution of BM alloying elements, such as Cr. The EDS analysis of point B in Figure 4b probably showed the high concentrations of Ni and Cr because of the formation of Cr-Ni-rich boride precipitates. The EDS analysis of point H showed the high value of Ni, and the EDS line that passed in Figure 5c at 0–2 µm indicated the high values of Ni and B; this area may be an Ni-B compound. In addition, the investigation of Ni-Cr-B ternary systems showed that eutectic phases and single-phase γ-solid solutions were formed from the remaining liquid phase at 1110 °C and 1096 °C based on the following phase transitions [36,37]:

According to eutectic reactions from the Cr-Ni-B liquid phase observed at 1110 °C.
*L → Ni*_3_*B + CrB*(1)

Finally, the liquid phase was transformed into a nickel-rich boride, chromium-rich boride, and eutectic γ- phase through a ternary eutectic reaction at about 1096 °C.
*L→ Ni*_3_*B + Ni*_2_*B + CrB + γ_eutectic_*(2)

(2)The Mo-rich boride binary eutectic and Ni-rich γ-phase (Ni-Mo-B) were formed based on the EDS spectrum of point C in Figure 4c and the EDS analysis (Table 2). These phases were formed by a combination of Mo from the BM (K > 1) and the MPD elements (K < 1) that remained in the liquid phase. According to the study of the Mo-Ni-B ternary phase diagram, Ni-Mo boride precipitates were formed at 1080 °C through the following phases [38]:


*L →*
*γ*
*+ NiMo*
_2_
*B*
_2_
*+ Ni*
_3_
*B*
(3)


(3)$Ni_{3}Si$ phases were possibly formed at 1040 °C based on the Ni-Si phase diagram in Figure 6b and the concentration of Si and Ni (EDS analysis of point D, I, and K and Figure 4d in this area). Moreover, γ-eutectic phase including $Ni_{3}Si$ was formed in this area because the Si content was more than its solubility. Oikawa et al. [39] reported that during cooling, excessive amounts of Si atoms were repelled from the γ-solid solution, so small cubic-shape precipitates were formed. Figure 5b shows that Si and Ni are present in high amounts, in contrast to other elements.

(4)Cr-Mo-rich compounds were observed in the joint centreline, considering the dissolution of BM alloying elements into the bonding area. The EDS analysis of point E showed the high concentrations of Cr and Mo. The EDS line of B (Figure 5) showed a high intensity in this area, indicating that it is super-saturated with B. According to the EDS analysis results and the clarification by Tojo et al. [40], the following phases were formed at 1000 °C during the cooling cycle:


*L → L + (Cr-Mo)*
_2_
*B*
(4)


(5)The EDS line in Figure 5 showed that in the 15–20 µm, B, Si, and Ni have high values, which may be Ni-Si-B compounds. The EDS data for point F (Figure 4f) and point L (Figure 6a) in Table 2 and Table 3 indicated that Ni-B-Si may be formed in these areas. Tokunaga et al. [41,42] reported similar results after studying Ni–Si–B ternary systems. *Ni*_2_*B*, *Ni*_3_*B* and *Ni*_6_*Si*_2_*B* phases may be formed at 850–990 °C by eutectic transformation from the remaining liquid phase from the last stage and during cooling process to room temperature, as shown in the following reaction:


*L → Ni*
_2_
*B + Ni*
_3_
*B + Ni*
_6_
*Si*
_2_
*B*
(5)


The two mechanisms that control the transformation during ISZ are as follows: the dissolution of the BM-alloying elements (i.e., Mo and Cr) and the diffusion of MPD atoms (i.e., B). Considering the mass transfer, the concentration of all BM alloying elements in the joining zone can be estimated by calculating the diffusion rates. The diffusion rate ($D_{diff.}$) can be determined by dividing the BM solute mass by the liquid mass:(6)$$D_{diff.}=\frac{M_{Tot}-M_{0}}{M_{Tot}}$$
where $M_{Tot}$ is the mass of the liquid phase and $M_{0}$ is the mass of the liquid phase before the dissolution of the BM alloying elements. Determining the liquid phase mass is impossible before and after dissolution; however, by knowing that the density of the BM and interlayer will remain similar, the dissolution rate can be calculated by the following equations:(7)$$D_{diss.}=1-\frac{W_{0}}{W_{max}}$$
where $W_{max}$ is the maximum width of the joining zone after the DB process and $W_{0}$ is the thickness of the primary interlayer. Therefore, the concentration of each alloy elements after dissolution can be determined through the following equation:(8)$$C_{diss.}=D_{diss.}\times C_{BM}+\left(1-D_{diss.}\right)\times C_{Int.}\;$$
where $C_{BM}$ and $C_{Int.}$ are the concentrations of the elements in the BM and interlayer before diffusion, respectively. Therefore, the following equation can be proposed for determining the contribution of diffusion during the IS step ($C_{diff.}$) in the total amount of all the elements in the bonding area ($C_{T}$):(9)$$C_{diff.}=C_{T}-C_{diss.}$$

Accordingly, $W_{0}$ and $W_{max}$ were assumed to be 75 and 151.18 µm, respectively (the measurement was conducted using Image J software). The dissolution rate in the joining zone remained at 0.5 for 30 min. $C_{T}$ (based on the EDS analysis from Table 2), $C_{diff.}$, and $C_{diss.}$ for the diffusion and dissolution elements in the centreline are listed in Table 4.

The negative values of $C_{diff.}$ for Cr and Mo in Table 2 may be due to the incomplete formation of ASZ or ISZ. The presence of Cr- and Mo-rich boride precipitates in ASZ prohibited the diffusion of Cr and Mo into the matrix. Moreover, hard and brittle boride precipitates were formed in ASZ due to the distribution factor of less than one K<1 and the poor solubility of B in Cr and Mo.

#### 3.1.3. Diffusion Affected Zone

A precipitation area was formed in DAZ by the diffusion of MPD atoms, such as B, into the BM. Figure 7a shows that this precipitation area had three different morphologies, such as (1) cubic, (2) needle, and (3) grain boundary, which were formed close to ISZ, beyond the cubic shape and into the grain boundary, respectively. According to the EDS analysis of points M, N and O in Figure 7b,c,e (Table 5), the concentrations of Cr, Ni, and Mo were higher than the solubility in the BM; as such, the formation of Cr, Ni, and Mo-rich boride precipitates is possible in this area. 

The majority of boride precipitates were formed during bonding at the bonding temperature. Table 5 shows that the DAZ does not have any Si in its chemical composition mainly because Si has lower diffusivity and higher solubility in Ni than in B; thus, Si cannot diffuse into the BM rapidly. In addition, given that B is a light element and is lighter than silicon, it has smaller diffusivity [43,44]. The formation of boride precipitates is mainly due to two dependent factors, namely, the penetration of B into the BM during the DB process and the presence of Mo in the BM, which can form borides. The diffusion of B into the BM leads to the formation of Cr- and Mo-rich boride precipitates close to the joining zone. This process might be due to the low solubility of B in Mo and Cr. DAZ had several phases based on the EDS analysis and the EDS line of B (Figure 7e); these phases can be summarised as follows: (1)Cubic precipitates: based on EDS data for point M, the concentrations of Mo and Cr were very high in the precipitated phases. The formation of Cr- and Mo-rich borides was highly probable due to the high concentration of B in these areas. Additionally, the EDS analysis spectra illustrated in Figure 7b showed the high concentrations of Cr and Mo and the presence of B.(2)Needle-shaped precipitates: the analysis of point N (Figure 7c) showed that this area had higher concentrations of Cr and B inside the grains and higher solubility than that for the matrix. As a result, this phenomenon led to the formation of Cr-rich boride precipitates.(3)Grain boundary precipitates: these precipitates were formed in the grain boundary and based on the EDS analysis of point O (Figure 7d) and EDS line of the B element. These areas are Mo-rich borides. The grain boundaries are favourable paths for the diffusion of atoms, such as MPD elements (i.e., B element), due to low atomic density.

### 3.2. Corrosion Study

#### 3.2.1. Polarisation Test

Figure 8 shows the cyclic polarisation plot from BM and DB samples. The details of the electrochemical parameters obtained from the cycle polarisation test are summarised in Table 6. The BM showed higher corrosion current density ($i_{corr}$) than that for the DB sample. Although it may seem that the joint region has a higher corrosion resistance than the BM. It’s attributed to computing the $i_{corr}$ in relation with the whole exposed area. However, at a microlevel point of view, the only areas that were corroded, were the ones that did not have a passive layer; therefore, at the joint region the corrosion would be localized but in reality the actual area is smaller, thus the true $i_{corr}$ is larger than the measured $i_{corr}$.

In the cyclic polarisation test, when the potential was scanned in the opposite direction, a hysteresis loop was formed because of the delay in the process of an existing pit becoming passive again [21]. Figure 6 shows the plot of cyclic polarisation as well as the position of $E_{corr}$, breakdown, and re-passive potentials ($E_{b}$ and $E_{rp}$, respectively). The potential in which the branching of the inverted anodic current cut the forward anodic scan was defined as $E_{rp}$. The width of the passive region - ($E_{b}-E_{corr}$)can be considered as a value which represents the corrosion resistance of a metal [45]. Other researchers defined the difference between $E_{b}$ and $E_{rp}$ as pitting corrosion sensitivity [46].

In Table 6, the $E_{b}-E_{corr}$ values were compared between BM and DB samples. When the passive region was wider (larger $E_{b}-E_{corr}$), the resistance against the pit formation was higher. The amount of $E_{b}-E_{corr}$ for BM (0.72 V) was approximately twice as high than that for the DB sample (0.30 V). These results indicated that Inconel 625 alloy has an acceptable corrosion behaviour, and the DB process can reduce the resistance of the alloy to local attacks in $Cl^{-}$-containing environments.

As shown in Table 6, the BM has a negative hysteresis loop and low $E_{b}-E_{rp}$ value (−0.38 *V*). Therefore, the existing pit can be passivated easily. The DB sample had larger positive hysteresis loop and higher $E_{b}-E_{rp}$ value (0.34 *V*). The large magnitude of $E_{b}-E_{rp}$ prohibited the repassivation of the pits, so the DB sample suffered from pitting during the direct scan. 

According to Figure 9 the pitting corrosion occurred in the joining zone. The polarisation results showed that the joining zone is the suitable area for the initiation of pitting. Pardo et al. [47] stated that the low concentration of Mo initiates local corrosion. The Mo deficiency in the joining zone, especially in ASZ, is shown in Figure 9d. 

As discussed previously, the ASZ area was formed in the centreline (Figure 3a) and had a microstructure including Cr-, Mo-, and Ni-rich borides; as such, around boride-rich regions, alloying elements, such as Cr and Mo, were depleted. Thus, in the ASZ phase, due to multiple and compacted precipitates, this area is not exposed to the corrosive solution. So corrosion attacks may not occur. However, in the ISZ zone, due to depletion of alloying elements as result lack of a passive layer or the formation of a weak passive layer and the lack of boride and silicide precipitation (according to Figure 3), corrosion attacks can occur. 

Figure 9 shows the depletion of Cr is less in the DAZ than in ASZ + ISZ; as such, needle-like and cubic boride precipitates in DAZ form passive layers by creating the diffusion paths for alloying elements, such as Mo and Cr. Zhang et al. [48] stated that the refined microstructure provides excess paths for diffusion to passivate the alloys. The fine precipitates in DAZ leading to better corrosion resistance compared with the large precipitates in ASZ could be due to the presence of fine precipitates with high boundaries, thereby facilitating the formation of a conducting passive layer. Thus, possibility of corrosion in this area would be lower [26].

Overall, Cr is the main component for conducting protective passive layers, leading to an increase in the general corrosion of the material by producing an adhesive chromium oxide layer on the metal surface. Nonetheless, in the case of damage (broken layer), the material is exposed to pitting corrosion. To solve this problem, Mo is used [47]. This element plays a remarkable role in the repassivation of the passive layer. The percentage of Mo (almost 9 wt%) in the alloy background is less than that of chrome (almost 22 wt%). Thus, in the present paper, both elements were removed from the background due to the DB process; as such, the volume of Mo depletion is higher, and as a result, the DB sample is very sensitive to pitting corrosion. 

#### 3.2.2. EIS test

The Nyquist charts of Inconel 625 and DB sample in 3.5 wt% NaCl solution at ambient temperature are shown in Figure 10a. The shape of the Nyquist curves was not ideal semicircles. The non-ideal semicircles are known as depressed semicircles, whose centers lie below the real impedance axis; this phenomenon is known as the dispersive effect [49]. The depressed semicircles can be caused by surface heterogeneity or represent the charge transfer controlling the corrosion behaviour [50].

An equivalent circuit is drawn and illustrated in Figure 10b to interpret the EIS plots [51]. In this circuit, $R_{s}$ is the resistance of the electrolyte, $C_{ox}$ is the capacitance of the oxide layer, $R_{ox}$ is the resistance of the oxide layer, $R_{ct}$ is the charge transfer resistance, and $C_{dl}$ is the capacitance of the double layer. A CPE element was used, whose impedance was calculated based on Equation (10), because the double layer formed on the electrode surface acts as a non-ideal capacitor, instead of using C as the ideal capacitor capacity.
(10)$$\;Z_{CPE}=\left(\frac{1}{Y_{0}}\right)[{\left(J\times 2\pi f_{max}{)}^{n}\right]}^{-1}$$

In this equation, $Y_{0}$ is the characteristic modulus, $J=\sqrt{-1}$, $f_{max}$ is the maximum AC frequency, and n is the phase transfer (1 > n > −1). The following equation was used for the calculation of double-layer capacitance.
(11)$$C_{dl}={\left(Y_{0}\times R_{ct}{}^{1-n}\right)}^{\frac{1}{n}}$$

The $R_{ox}-C_{ox}$ circuit was observed in high- and medium-frequency regions. $R_{ox}$ is the ionic current resistance, and $C_{ox}$ indicates the capacity of the passive film. The $R_{ct}-C_{dl}$ circuit can be observed at low frequencies. The results of the EIS data fitting are summarised in Table 7. The values of $n_{ox}$ and $n_{dl}$ were within 0.84–0.88 and 0.76–0.82, respectively, indicating that the CPEs play the role of a capacitor [52]. However, if ‘n’ is smaller than 1, then a deviation of the pure capacitor in the circuit components exists, which typically occurs for oxide layers. The value of n depends on the non-uniform distribution of the current and the surface roughness, which means lower values of n indicate a rougher electrode surface [53]. Therefore, the passive film formed on the surface of the joint area was defective and corrosion occurred on the surface of the specimens.

The polarisation resistance of $R_{p}=R_{ox}+R_{ct}$ was used to compare the corrosion behaviour of the samples. Using the overall charge transfer resistance is better instead of $R_{ox}$ or $R_{ct}$ individually [21]. A larger value of $R_{p}$ indicates better general corrosion resistance. Therefore, the Inconel 625 sample ($R_{p}$= 3.16 $M\Omega \;cm^{2}$) exhibited lower corrosion resistance compared with the DB sample ($R_{p}$= 5.38 $M\Omega \;cm^{2}$). Table 7 shows that the amount of $C_{ox}$ increased after diffusion brazing. According to the Helmholtz model, C is defined as Equation (12) [54]:(12)$$C=\frac{\varepsilon^{{}^{\circ}}\varepsilon }{d}A$$
where $\varepsilon^{{}^{\circ}}$ is the vacuum dielectric constant, ε is the local dielectric constant, A is the exposed surface to the electrode, and d is the thickness of the electric double layer.

Therefore, the increase in $C_{ox}$ may be due to decrease in the thickness of the passive layer, which was formed in the joining zone. Therefore, although the $R_{p}$ of the joining zone is higher than the $R_{p}$ of the BM, in reality, the actual area exposed to corrosive solution is considered less than 0.1 $cm^{2}$ due to the formation of precipitates and also less thickness of the passive layer formed in the joining zone; consequently, it is weaker than the passive layer formed in the base metal. It can thus be concluded that the corrosion resistance of the joint area is in fact less than BM. These results agree with the polarisation test.

## 4. Conclusions

During diffusion brazing with the B-Si-Cr-Fe-Ni filler metal, the formation of intermetallic compounds is unavoidable when the gap size is wide. Controlling and determining the distribution and size of these compounds is the best method for producing a strong brazed bond. Regarding the effects of diffusion brazing on the microstructure and corrosion behaviour at 1120 °C for 30 min, the following highlighted results were obtained:

ISZ was formed due the dissolution of BM elements, such as Mo, Cr, Nb, and Fe, into the joining zone. The joining zone has a Ni-rich γ-solid solution composition. According to the microstructure results, this area is the same in the base metal but has fewer alloying elements, such as Mo, Cr, and Nb. 

In the ASZ, the following chemical reaction occurred during the cooling cycle from 1100 °C to 850 °C:$$L\;\left(remaining\;liquid\;during\;the\;cooling\;–cycle\right)\to \gamma eutectic+\left(Ni_{3}B,\;Ni_{2}B\right)+NiMo_{2}B_{2}+{\left(Cr-Mo\right)}_{2}B+CrB+Ni_{6}Si_{2}B+Ni_{3}Si$$

The formation of these compounds leads to incomplete ISZ and the heterogeneity of the joining zone. According to the microstructure of Ni–boride and Ni–Si–B intermetallic interactions in the ASZ area, they are the weak micro-constituents in the joining zone. Therefore, these compounds are unstable.

As a result of the B diffusion into the base metal close to the joining zone, DAZ with a cubic morphology was formed. The morphologies of the phases in DAZ are cubic and needle shaped and the grain boundary by Cr-rich, Cr- and Mo-rich, and Mo-rich boride, respectively.

The EIS and cyclic polarisation test results showed that the Inconel 625 alloy has an acceptable corrosion behaviour in 3.5 wt% NaCl solution. The DB process reduces the passive region and then increases the sensitivity of the joining zone to pitting corrosion.

In the ASZ zone, due to multiple and compacted precipitates, corrosion attacks may not occur. In the ISZ zone, due to the depletion of alloying elements and also lack of precipitation, corrosion attack can occur. In the DAZ, due to the presence of fine precipitates with high boundaries, thereby facilitating the formation of a conducting passive layer, the possibility of corrosion in this area is lower.

## Figures and Tables

**Figure 1 materials-16-05072-f001:**
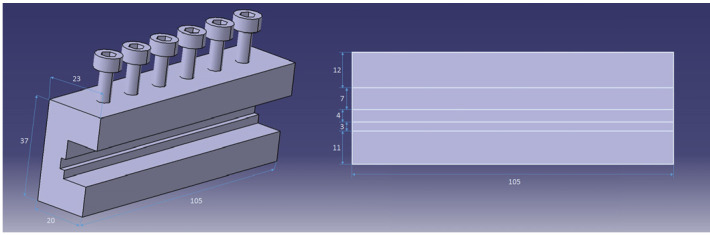
The schematic of the holding fixture (mm).

**Figure 2 materials-16-05072-f002:**
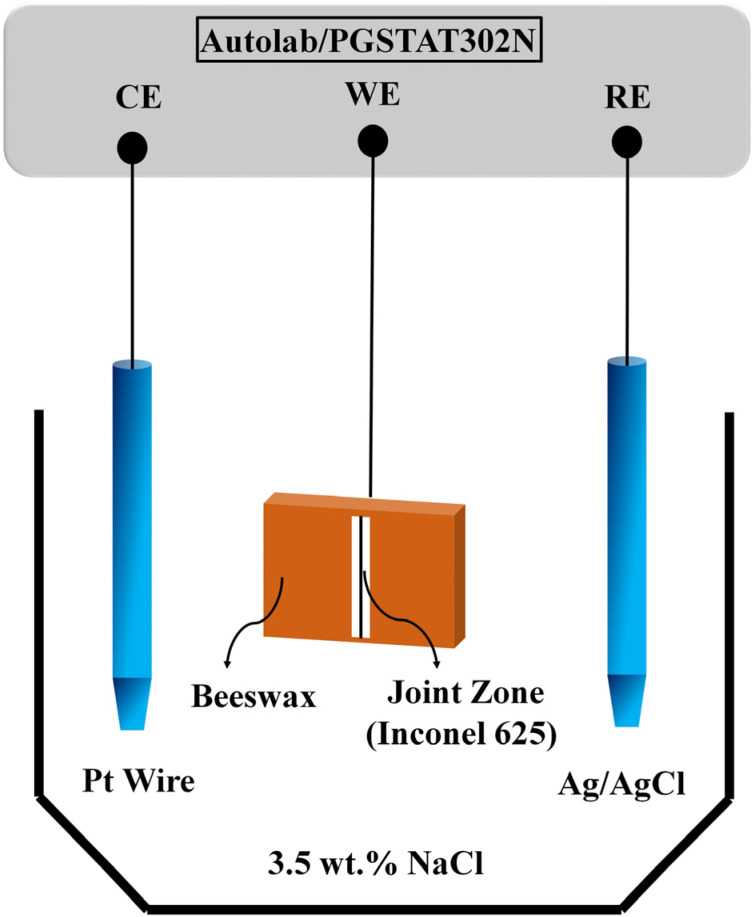
The schematic of the electrochemical test.

**Figure 3 materials-16-05072-f003:**
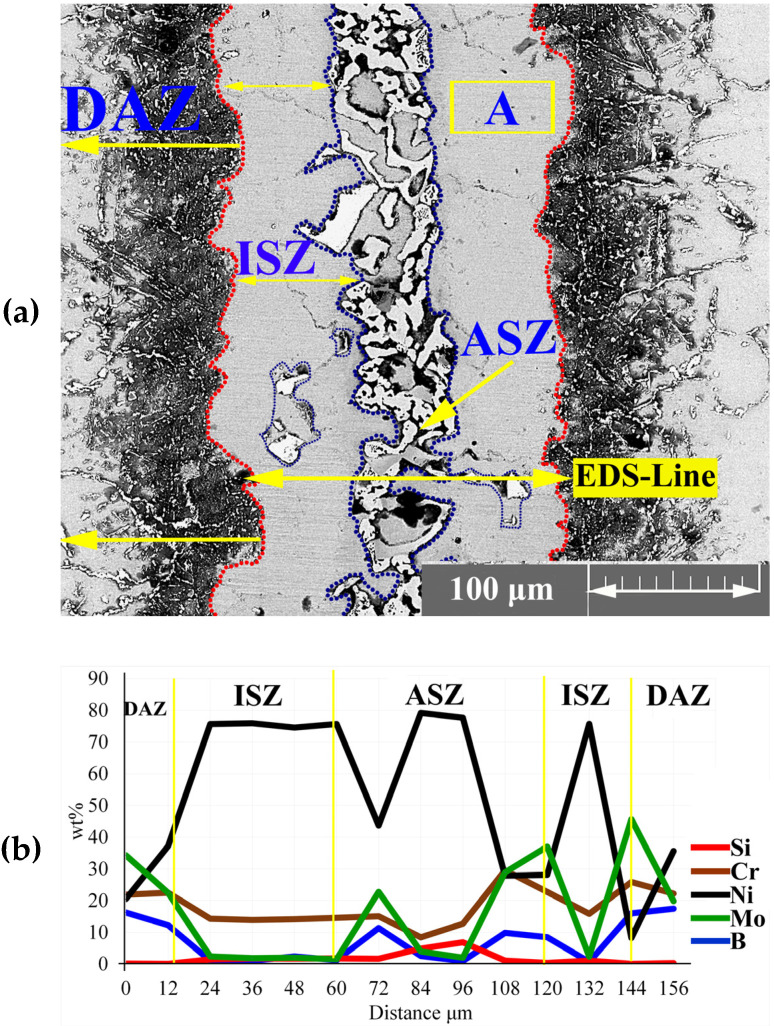
EDS analysis of joining zone; (**a**) FESEM image (Detector: SE) of the joining zone; and (**b**) EDS line scanning results.

**Figure 4 materials-16-05072-f004:**
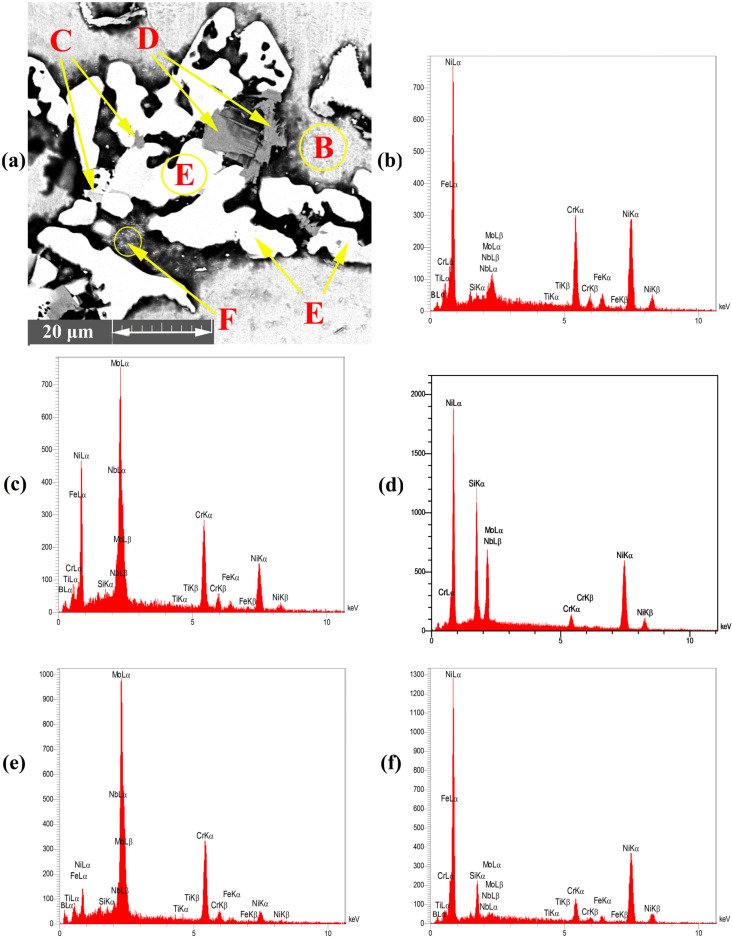
EDS point analysis of ASZ region; (**a**) FESEM image (Detector: SE) of ASZ region, (**b**) point B, (**c**) point C, (**d**) point D, (**e**) point E, and (**f**) point F.

**Figure 5 materials-16-05072-f005:**
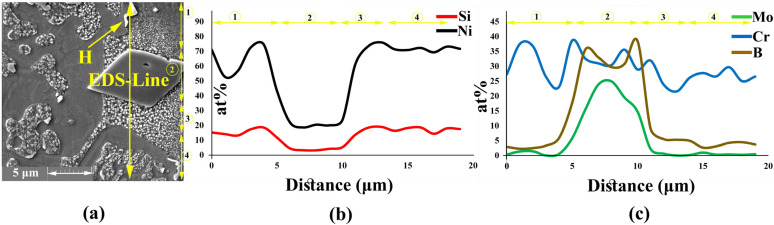
EDS analysis at ASZ; (**a**) FESEM image (Detector: SE); (**b**) and (**c**) EDS analysis of eutectic structure.

**Figure 6 materials-16-05072-f006:**
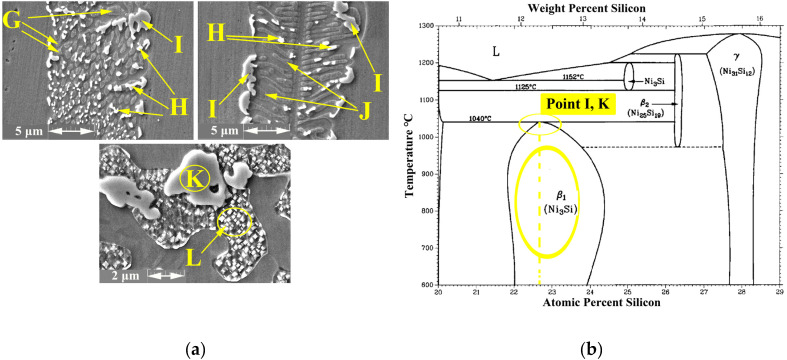
EDS analysis; (**a**) FESEM Image (Detector: SE) of ASZ region and (**b**) Ni-Si diagram phase of point I and K.

**Figure 7 materials-16-05072-f007:**
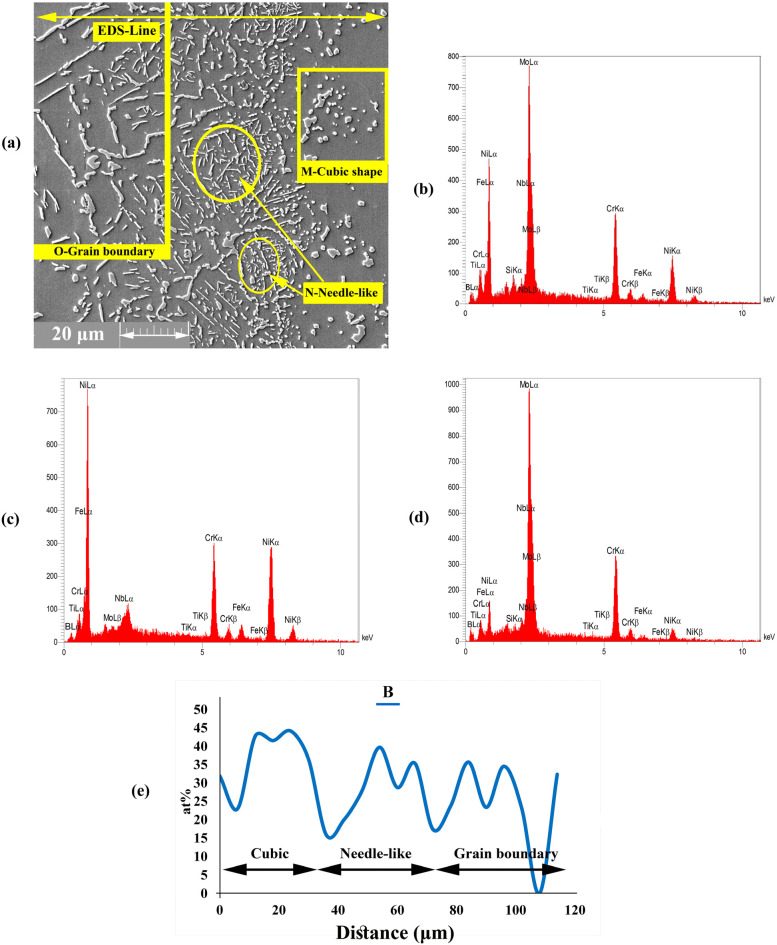
EDS analysis of DAZ microstructure; (**a**) FESEM images (Detector: SE) of DAZ region; and EDS spectrum of (**b**) M region, (**c**) N region, (**d**) O region, and (**e**) EDS line scanning of B at 1120 °C.

**Figure 8 materials-16-05072-f008:**
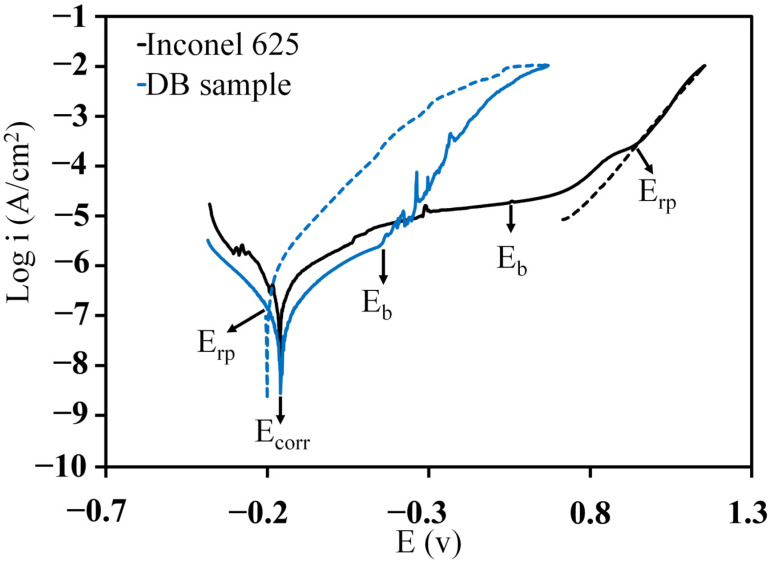
Cyclic polarization curves for Inconel 625 and DB sample in 3.5 wt% NaCl solution (continuous line indicates the direction of flow scanning and dotted line indicates the reverse flow scan).

**Figure 9 materials-16-05072-f009:**
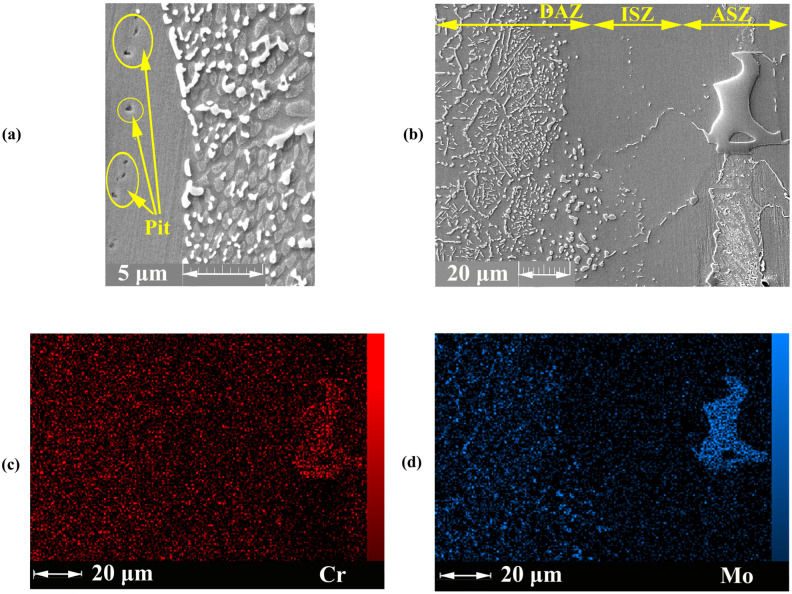
FESEM image (Detector: SE) of the joining zone; (**a**) ASZ region, EDS mapping analysis of the joining zone; (**b**) FESEM image; (**c**) Mo; and (**d**) C.

**Figure 10 materials-16-05072-f010:**
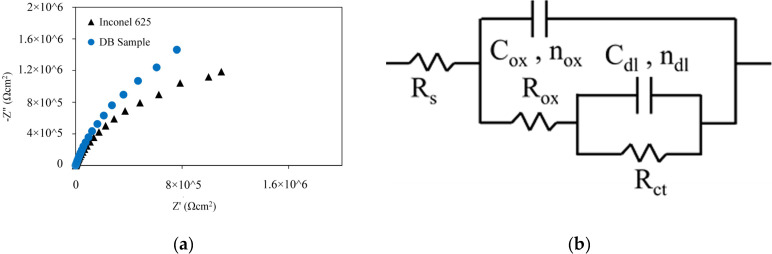
EIS results of Inconel 625 and DB sample in 3.5 wt% NaCl solution; (**a**) Nyquist curves and (**b**) electrical equivalent circuit.

**Table 1 materials-16-05072-t001:** The chemical composition of Inconel 625 and MBF-20.

At.%	Ni	Cr	Fe	Mo	Nb	C	Mn	Al	Ti	Co	B	Si
Inconel 625	58.95	25.46	5.08	5.84	2.38	0.99	0.43	0.87	0.49	0.87	-	-
MBF-20	67.55	6.96	2.34	-	-	0.25	-	-	-	-	14.8	8.1

**Table 2 materials-16-05072-t002:** Chemical composition (at. %) indicated in Figure 3a and Figure 4a.

Point	Ni	Mo	Cr	Nb	Si	Fe	Phase Predicted
Point A	74.6	5.43	11.21	2.69	3.67	2.25	γ-solid solution
Point B	60.10	3.28	32.43			2.11	Ni, Cr-rich boride
Point C	61.15	34.73				1.87	Ni, Mo-rich boride
Point D	75.23				21.81	2.96	Ni-rich silicide
Point E	11.62	36.12	48.63			1.63	Mo, Cr-rich boride
Point F	76.35	-	7.44		10.65	2.58	γ-eutectic Ni-Si-B

**Table 3 materials-16-05072-t003:** Chemical composition (at. %) indicated in Figure 5a.

Point	Ni	Cr	Mo	Nb	Si	phases
Point G	76.49	8.95	3.92	2.48	8.16	Matrix
Point H	73.08	14.09	2.45	4.65	5.62	Ni-rich Boride
Point I	64.18	3.87	3.52	5.31	23.18	Ni-rich silicide
Point J	81.79	7.37	5.47	1.13	4.24	Matrix
Point K	65.78	3.71	4.17	1.75	24.19	Ni-rich silicide
Point L	72.96	7.12	2.01	1.05	17.76	Ni-Si-rich boride

**Table 4 materials-16-05072-t004:** EDS analysis (at %) of alloying elements at the joining zone of the DB bonded for 30 min.

Concentration	Ni	Cr	Mo	Fe	Nb
** $C_{BM}$ **	58.95	24.07	5.84	5.08	2.38
** $C_{Int.}$ **	67.55	6.99	0	2.34	0
** $C_{diss.}$ **	63.24	15.53	2.92	3.71	1.19
** $C_{diff.}$ **	11.04	−3.25	−0.91	0.04	0.23
** $C_{T}$ **	73.94	12.28	2.01	3.75	1.42

**Table 5 materials-16-05072-t005:** Chemical composition (at %) indicated in Figure 6a.

Point	Ni	Mo	Cr	Nb	Si	Fe	Phase Predicted
Point M	20.43	32.23	42.28	-	-	2.31	Cr, Mo-rich boride
Point N	30.50	8.93	58.31	-	-	1.84	Cr-rich boride
Pont O	10.41	56.23	23.34	-	-	1.38	Mo-rich boride

**Table 6 materials-16-05072-t006:** Potentiodynamic polarization for Inconel 625 and DB samples in 3.5 wt.% NaCl solution.

**Sample**	** $i_{corr}$ ** ** $\left(\mu A/cm^{2}\right)$ **	$E_{corr}$ (*V*)	$E_{b}$ (*V*)	$E_{rp}$ (*V*)	$E_{b}-E_{corr}$ (*V*)	$E_{b}-E_{rp}\;(V)$
**Inconel 625**	0.330	−0.16	0.56	0.94	0.72	−0.38
**DB sample**	0.132	−0.15	0.15	−0.19	0.30	0.34

**Table 7 materials-16-05072-t007:** EIS parameters obtained from Inconel 625 samples and DB samples in 3.5 wt% NaCl.

Sample	$R_{s}\;$ $\left(\Omega \;cm^{2}\right)$	$R_{ox}$ $\left(K\Omega \;cm^{2}\right)$	$C_{ox}\;$ $\left(µF\;cm^{-2}\right)$	$n_{ox}$	$R_{ct}$ $\left(M\Omega \;cm^{2}\right)$	$C_{dl}\;$ $\left(µF\;cm^{-2}\right)$	$n_{dl}$	$R_{p}$ $\left(M\Omega \;cm^{2}\right)$	hi₋squared $\left(\times 10^{-3}\right)$
Inconel 625	29.8	62.82	3.53	0.84	3.10	0.003	0.76	3.16	1.4
DB sample	28.75	78.25	4.95	0.88	5.30	1.10	0.82	5.38	0.7

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
