# Peer review of "Effect of MBF-20 Interlayer on the Microstructure and Corrosion Behaviour of Inconel 625 Super Alloy after Diffusion Brazing"

_materials, 2023, doi:10.3390/ma16145072_

Round 1
Reviewer 1 Report
1. It is recommended to mark the location of the δ and titanium carbide phase in Figure 3(a).
2. In order to further illustrate the distribution and formation of compounds caused by the diffusion of alloying elements during brazing, should different brazing times be better?
3. Is "Fig.2a" in line 170 correct?
4. Is it accurate to use only EDS analysis for the determination of intermetallic compounds? A qualitative analysis should be performed with RXD first, followed by the use of EDS to determine the location.
Author Response
Thank you very much for your message dated Jun 14, 2022 and for the comments and suggestions made by the respectful reviewers to improve the manuscript entitled " Effect of MBF-20 Interlayer on the Microstructure and Corrosion Behaviour of Inconel 625 Super alloy after Diffusion Brazing " (Ref: materials-1732852).
As below, on behalf of my co-authors, I would like to clarify some of the points raised by the Reviewers. And we hope the Reviewers and the Editor will be satisfied with our responses to the ‘comments’ and the revisions for the original manuscript. Also, the corrections are listed as follows.
Reviewer 1
- It is recommended to mark the location of the δ and titanium carbide phase in Figure 3(a).
Answer: If we want to show these phases in DB samples, all joining surface may be over-etched. Since, it takes a long time to appear grain and grain boundary than the etch time for detecting joining zone. So this could be harmful for detecting new precipitation after DB.
- In order to further illustrate the distribution and formation of compounds caused by the diffusion of alloying elements during brazing, should different brazing times be better?
Answer: I have studied a wide variety of time bonding (between 30 t0 120 min.), and know that we could detect all phase in the best formation condition in 30 min. bonding time
- Is "Fig.2a" in line 170 correct?
Answer: It has amended.
- Is it accurate to use only EDS analysis for the determination of intermetallic compounds? A qualitative analysis should be performed with RXD first, followed by the use of EDS to determine the location.
Answer: XRD analysis could not detected the phase of B element because of its less amount.
Reviewer 2 Report
1) The introduction part should be improved to illustrate a comprehensive references.
2) The electrochemical experiments described in the text is not suitable for characterize the corrosion behaviour of the electrode. corrosion state of th electrode won't be really stable when the corrosion potential becomes stable at the first time . The corrosion behaviour should be a continuous monitoring experiments.
3)The corrosion behaviour is a mixture of the three zones. It si better to clarify the corrosion behaviour of different zones.
4) The author didn't illustrate the experiment clearly.
5)The Figure should be interpreted more clearly, such as Figure 1.
6) Why is only 625 and DB samples tested?
Author Response
Thank you very much for your message dated Jun 14, 2022 and for the comments and suggestions made by the respectful reviewers to improve the manuscript entitled " Effect of MBF-20 Interlayer on the Microstructure and Corrosion Behaviour of Inconel 625 Super alloy after Diffusion Brazing " (Ref: materials-1732852).
As below, on behalf of my co-authors, I would like to clarify some of the points raised by the Reviewers. And we hope the Reviewers and the Editor will be satisfied with our responses to the ‘comments’ and the revisions for the original manuscript. Also, the corrections are listed as follows.
1) The introduction part should be improved to illustrate a comprehensive references.
Answer: It has already done.
2) The electrochemical experiments described in the text is not suitable for characterize the corrosion behaviour of the electrode. corrosion state of th electrode won't be really stable when the corrosion potential becomes stable at the first time. The corrosion behaviour should be a continuous monitoring experiments.
Answer: Answer: Thanks for the suggestion. Corrosion potentials of DB and BM samples were stabilized after approximately 20 minutes immersion in 3.5 wt% NaCl. However, electrochemical tests were performed after 60 minutes to ensure more complete corrosion potential stability. Also, before starting each test based on the applied procedure, the test will not be run until dE / dt <10-6 V. In addition, due to the high corrosion resistance of Inconel 625, it is expected that the corrosion behaviour of the sample does not change much over time. Also, other researchers in similar works have limited the study of corrosion behavior to the initial times of immersion.
https://doi.org/10.1016/j.jmapro.2019.04.029
https://doi.org/10.1016/j.jmapro.2018.11.025
https://doi.org/10.1007/s12540-020-00625-9
https://doi.org/10.1007/s12540-020-00671-3
https://doi.org/10.1007/s11661-016-3837-4
3)The corrosion behaviour is a mixture of the three zones. It is better to clarify the corrosion behaviour of different zones.
Answer: It has been checked and amended (lines:347-371).
4) The author didn't illustrate the experiment clearly.
Answer: The experiments were described in more detail. Below is the procedure in this study:
1-Selet the raw materials ( brand, and purity).
2- DB process.
3-list characterization tests.
- Microstructuer
- EDS/FESEM
- Corrosion test
5)The Figure should be interpreted more clearly, such as Figure 1.
Answer: We suffered many problems to make a fixture in this study. Because the samples have to fix on each other to an interlayer was made. Furthermore, due to the high-temperature process, the fixture is manufactured of stainless steel in an attempt to prevent deformation during the bonding process. So, a homemade stainless-steel fixture (Fig. 1) was used to grab the samples inside the furnace.
6) Why is only 625 and DB samples tested?
Answer: This work studied the microstructure of the joining zone for Inconel 625 superalloy after diffusion brazing (1120 °C for 30 minutes). To the best of our knowledge, no research has reported on the microstructure analysis of this alloy and the processing condition mentioned. The formation of borides and silicates in the joining zone were observed, and finally, a correlation was proposed between microstructure, corrosion behavior, and bonding temperature. Consequently, we decided to study only samples before and after the bonding process, in an attempt to investigate ultimately all of the compounds, which is formed during diffusion brazing in the joining zone.
Reviewer 3 Report
The present paper includes interesting results regarding the microstructure and corrosion behavior of the diffusion bonding zone. However, some comments can be made.
3.1. Microstructure characterization
a) Within the whole manuscript the authors used the abbreviation BM without to explain the mean of this; the authors should correct this lack.
b) at p. 4, lines 130-131, the authors wrote: „ …..the dissolution of the BM and inter-diffusion between BM and interlayer, leading to the formation of boride and silicate precipitates…..”; the authors should explain how silicate (silicon, oxygen and one or more metals) precipitates is possible to form, or the formed precipitates are silicides ( a binary compound of silicon and a more electropositive element).
c) at p. 5, lines 148-149, the authors introduce the quantity ”distribution coefficient (K)”, but no description of this there is in the manuscript; the authors should make a brief description of this quantity.
d) Using binary and ternary phase diagrams, as well as Eds analysis, the authors showed, in sections ”Athermal solidification zone” and ”Diffusion Affected Zone”, that different reactions are possible to produce different borides and silicides, but without they to indicate clearly which are the phases formed; to identify more preciously the phases formed in different zone of diffusion bonding area, the XRD analysis of this area is required, least;
e) In Table 2 and Figure 4a, the chemical compositions and phase morphology of point D and point F are similar, but the authors associated point D with Ni-rich silicides, and point F with gamma eutectic Ni-Si-B; the authors should explain supplementary this association.
3.2.1. Polarization test
f) at p. 12, line 305, the authors wrote that ” According to Table 6, the corrosion potential (Ecorr) of the DB sample was equal to Ecorr = -0.15 V, which was greater than Ecorr = -0.16 V for the BM.”; for polarization tests the measured values of Ecorr in replicates commonly varies in a 20 mV range, so that the authors should sustain this statement using repeated measurements of the Ecorr values and checking the statistical hypothesis of the mean value equality, otherwise it cannot be sustained.
g) Having in view the important differences between microstructure of the three zones of diffusion bonding area, the corrosion mechanism is more complicated; there are differences between the corrosion potential of the each phase, so that different galvanic microcell is possible to form; also, the values of corrosion current density (icorr) is computed in relation with whole exposed area, but, at micro level, only certain zones from diffusion bonding are corroded, namely, the real area is smaller; thus, the value of icorr corresponding to the diffusion bonding zone is higher (a smaller value of the corroded area); from point of view of corrosion resistance, the occurring of pitting corrosion in diffusion bonding is more relevant in comparison with values of Ecorr and icorr; the authors should revise this section.
3.2.2. EIS test
h) regarding the EIS test, the authors should discuss the results having in view the complex microstructure of the bonding zone, the microstructure of the base matrix (BM) and differences in electrochemical behavior of each phase; is possible to form a passive film for each phase? and which is the oxides formed in correlation with each microstructural phase?
Author Response
Thank you very much for your message dated Jun 14, 2022 and for the comments and suggestions made by the respectful reviewers to improve the manuscript entitled " Effect of MBF-20 Interlayer on the Microstructure and Corrosion Behaviour of Inconel 625 Super alloy after Diffusion Brazing " (Ref: materials-1732852).
As below, on behalf of my co-authors, I would like to clarify some of the points raised by the Reviewers. And we hope the Reviewers and the Editor will be satisfied with our responses to the ‘comments’ and the revisions for the original manuscript. Also, the corrections are listed as follows.
The present paper includes interesting results regarding the microstructure and corrosion behaviour of the diffusion bonding zone. However, some comments can be made.
3.1. Microstructure characterization
- Within the whole manuscript the authors used the abbreviation BM without to explain the mean of this; the authors should correct this lack.
Answer: It has amended.
- b) at p. 4, lines 130-131, the authors wrote: „ …..the dissolution of the BM and inter-diffusion between BM and interlayer, leading to the formation of boride and silicate precipitates…..”; the authors should explain how silicate (silicon, oxygen and one or more metals) precipitates is possible to form, or the formed precipitates are silicide’s (a binary compound of silicon and a more electropositive element).
Answer: Unfortunately, we used the wrong vocabulary. The correct is Nickel silicide (Ni3Si).
- c) at p. 5, lines 148-149, the authors introduce the quantity ”distribution coefficient (K)”, but no description of this there is in the manuscript; the authors should make a brief description of this quantity.
Answer: K= is distribution coefficient (K) of an element, where CS and CL are the concentration of the solute in the solid and melt, respectively. So, if the K for elements is less than one, they tend to stay in the melt, and melt can be rich from them. Therefore, for K>1, the concentration of these elements in front of interface can increase due to the rejection of alloying element from the solid.
- d) Using binary and ternary phase diagrams, as well as Eds analysis, the authors showed, in sections ”Athermal solidification zone” and ”Diffusion Affected Zone”, that different reactions are possible to produce different borides and silicides, but without they to indicate clearly which are the phases formed; to identify more preciously the phases formed in different zone of diffusion bonding area, the XRD analysis of this area is required, least;
Answer: XRD analysis could not detected the phase of B element because of its less amount.
- e) In Table 2 and Figure 4a, the chemical compositions and phase morphology of point D and point F are similar, but the authors associated point D with Ni-rich silicides, and point F with gamma eutectic Ni-Si-B; the authors should explain supplementary this association.
Answer: In Fig.4, the analysis of point f was wrong, and we replace it with a new and correct analysis. The different of point “f and D” are in the amount of B elements in the chemical composition (Incomplete sentence) . So because of that the EDS analysis cannot show the accurate amount of B in the precipitations, we used the its qualitative quantity due to the fig.4f.
3.2.1. Polarization test
- f) at p. 12, line 305, the authors wrote that ” According to Table 6, the corrosion potential (Ecorr) of the DB sample was equal to Ecorr = -0.15 V, which was greater than Ecorr = -0.16 V for the BM.”; for polarization tests the measured values of Ecorr in replicates commonly varies in a 20 mV range, so that the authors should sustain this statement using repeated measurements of the Ecorr values and checking the statistical hypothesis of the mean value equality, otherwise it cannot be sustained.
Answer: This research just presented a part of our investigation of TLP process including effect of mechanical properties, microstructure and corrosion behaviours of Inconel 625 joint region.
https://doi.org/10.1016/j.jmapro.2019.01.019
https://doi.org/10.1016/j.jmapro.2020.02.005
https://doi.org/10.1016/j.jmrt.2020.07.015
As result almost the same corrosion behaviour of TLP samples were observed. Also, the samples with precipitated at joint region exhibited more positive corrosion potential rather than BM which published in previous researches., Other researchers have also similar results as follows.
https://doi.org/10.1007/s11661-016-3837-4
https://doi.org/10.1007/s12540-020-00625-9
In addition, in the current samples, the difference in corrosion potential (Ecorr) between the samples is 100 mV, which is remarkable despite of errors consideration. However, if reviewer opinion is to remove the results, the authors will Amite those which are mentioned.
- Having in view the important differences between microstructure of the three zones of diffusion bonding area, the corrosion mechanism is more complicated; there are differences between the corrosion potential of the each phase, so that different galvanic microcell is possible to form; also, the values of corrosion current density (icorr) is computed in relation with whole exposed area, but, at micro level, only certain zones from diffusion bonding are corroded, namely, the real area is smaller; thus, the value of icorr corresponding to the diffusion bonding zone is higher (a smaller value of the corroded area); from point of view of corrosion resistance, the occurring of pitting corrosion in diffusion bonding is more relevant in comparison with values of Ecorr and icorr; the authors should revise this section.
Answer: It has been checked and amended (line:347-371).
3.2.2. EIS test
- Regarding the EIS test, the authors should discuss the results having in view the complex microstructure of the bonding zone, the microstructure of the base matrix (BM) and differences in electrochemical behaviour of each phase; is possible to form a passive film for each phase? and which is the oxides formed in correlation with each microstructural phase?
Answer: We thank the reviewer for this comment. According to MAP-EDS results, ASZ and ISZ phases depletion from alloying elements are more than DAZ phase. Moreover, according to precipitates from in ASZ, almost completely cover the surface while in DAZ phase the precipitates form is finer and well distributed. So, ASZ phase will not be exposed to the corrosive solution due to multiple and compacted precipitates ;thereby; corrosion attacks may not occur. However, in ISZ zone due to depletion of alloying elements lack of passive layer or formation of weak passive layer and also lack of borides and silicates precipitation (according to fig. 3) corrosion attack would be occurred. In DAZ zone due to presence the fine precipitates, they can facilitate the diffusion of the alloying elements and increase the formation of passive layer possibility. Thus, possibility of corrosion in this area would be lower. The formed oxides are shown in tables 2,3 and 5.
Round 2
Reviewer 2 Report
The languag should be revised thoroughly.
Author Response
Dear Dr. Teezy Rong,
Thank you very much for your message dated Jul 1, 2022 and for the comments and suggestions made by the respectful reviewers to improve the manuscript entitled " Effect of MBF-20 Interlayer on the Microstructure and Corrosion Behaviour of Inconel 625 Super alloy after Diffusion Brazing " (Ref: materials-1732852).
Attached please find the electronic file of the revised manuscript in which all suggestions made by the reviewers are justified and the manuscript was amended and improved accordingly. Follows are the answer to reviewers’ comments.
Editor and Reviewer Comments:
Reviewer 1
1 In Tables 2 and 3, as well as p. 7, line 207 etc, the authors should replace “silicate” with “silicide”.
Answer: We thank the reviewer for this comment. It has been checked and modified.
- The authors should argue and explain supplementary the association between point D with Ni-rich silicides, and point F with gamma eutectic Ni-Si-B, although the chemical compositions and phase morphology (microstructure image Fig. 4a) are similar; the authors answered that they changed the chemical composition of F point in Table 2, but in Table 2 the chemical compositions are the same.
Answer: Thank you for your comment. All chemical composition of points were reviewed again, and the correct values for point "F" were tabulated in Table 2
3.Is "Fig.2a" in line 170 correct?
Answer: It has amended.
4The difference between Ecorr is only 10 mV (-0.160 V (-160 mV) and – 0,150 V (-150 mV)).
Answer: Ù… We thank the reviewer for this comment. It has been checked and modified.
All changes made on the revised manuscript are highlighted in red. We hope that the revised manuscript is now suitable for publication in the Journal of Materials.
With my warmest regards ,
Armin Rajabi, Assistant Professor
Department of Mechanical and Manufacturing Engineering, Faculty of Engineering and Built Environment, Universitiy Kebangsaan Malaysia, UKM, Bangi 43600, Selangor, Malaysia;
Reviewer 3 Report
The authors should have in view the following comments:
a) In Tables 2 and 3, as well as p. 7, line 207 etc, the authors should replace “silicate” with “silicide”.
b) The authors should argue and explain supplementary the association between point D with Ni-rich silicides, and point F with gamma eutectic Ni-Si-B, although the chemical compositions and phase morphology (microstructure image Fig. 4a) are similar; the authors answered that they changed the chemical composition of F point in Table 2, but in Table 2 the chemical compositions are the same.
c) The difference between Ecorr is only 10 mV (-0.160 V (-160 mV) and – 0,150 V (-150 mV)).
Author Response
Dear Editor,
Thank you very much for your message dated Jul 1, 2022 and for the comments and suggestions made by the respectful reviewers to improve the manuscript entitled " Effect of MBF-20 Interlayer on the Microstructure and Corrosion Behaviour of Inconel 625 Super alloy after Diffusion Brazing " (Ref: materials-1732852). The lists below are the responses to each comment.
Editor and Reviewer Comments:
Reviewer 1
1 In Tables 2 and 3, as well as p. 7, line 207 etc, the authors should replace “silicate” with “silicide”.
Answer: We thank the reviewer for this comment. It has been checked and modified.
- The authors should argue and explain supplementary the association between point D with Ni-rich silicides, and point F with gamma eutectic Ni-Si-B, although the chemical compositions and phase morphology (microstructure image Fig. 4a) are similar; the authors answered that they changed the chemical composition of F point in Table 2, but in Table 2 the chemical compositions are the same.
Answer: Thank you for your comment. All chemical composition of points were reviewed again, and the correct values for point "F" were tabulated in Table 2
3.Is "Fig.2a" in line 170 correct?
Answer: It has amended.
4The difference between Ecorr is only 10 mV (-0.160 V (-160 mV) and – 0,150 V (-150 mV)).
Answer: Ù… We thank the reviewer for this comment. It has been checked and modified.
All changes made on the revised manuscript are highlighted in red. We hope that the revised manuscript is now suitable for publication in the Journal of Materials.
With my warmest regards ,
Armin Rajabi, Assistant Professor
Department of Mechanical and Manufacturing Engineering, Faculty of Engineering and Built Environment, Universitiy Kebangsaan Malaysia, UKM, Bangi 43600, Selangor, Malaysia;